# Trends in Online Search Activity and the Correlation with Daily New Cases of Monkeypox among 102 Countries or Territories

**DOI:** 10.3390/ijerph20043395

**Published:** 2023-02-15

**Authors:** Min Du, Chenyuan Qin, Wenxin Yan, Qiao Liu, Yaping Wang, Lin Zhu, Wannian Liang, Min Liu, Jue Liu

**Affiliations:** 1Department of Epidemiology and Biostatistics, School of Public Health, Peking University, Beijing 100191, China; 2Center for Primary Care and Outcomes Research, School of Medicine, Center for Health Policy, Freeman Spogli Institute for International Studies, Stanford University, Stanford, CA 94305-2004, USA; 3Vanke School of Public Health, Tsinghua University, Beijing 100084, China; 4Global Center for Infectious Disease and Policy Research & Global Health and Infectious Diseases Group, Peking University, Beijing 100191, China; 5Key Laboratory of Reproductive Health, National Health and Family Planning Commission of the People’s Republic of China, Beijing 100191, China

**Keywords:** monkeypox, online search activity, emergency, disparities, public health emergency of international concern

## Abstract

Research assessing the trend in online search activity on monkeypox (mpox) and the correlation with the mpox epidemic at the global and national level is scarce. The trend of online search activity and the time-lag correlations between it and daily new mpox cases were estimated by using segmented interrupted time-series analysis and Spearman correlation coefficient (rs), respectively. We found that after the declaration of a Public Health Emergency of International Concern (PHEIC), the proportion of countries or territories with increasing changes in online search activity was lowest in Africa (8.16%, 4/49), and a downward trend in online search activity was highest in North America (8/31, 25.81%). The time-lag effect of global online search activity on daily new cases was significant (rs = 0.24). There were eight countries or territories with significant time-lag effect; the top three countries or territories were Brazil (rs = 0.46), United States (rs = 0.24), and Canada (rs = 0.24). Interest behavior in mpox was insufficient, even after the declaration of PHEIC, especially in Africa and North America. Online search activity could be used as an early indicator of the outbreak of mpox at the global level and in epidemic countries.

## 1. Introduction

Recently, a multiple-country mpox outbreak has attracted global public attention. Beginning with a child from the Democratic Republic of the Congo in 1970 as the first human mpox case, this sporadic zoonosis was caused by two variants of orthopoxvirus, namely, the Central African clade (clade I) and the West African clade (clade II), and began to spread in rural rainforest villages of western and central Africa [1,2]. The symptoms of mpox include skin rash, fever, intense headache, swelling of lymph nodes, back pain, muscle aches, and lack of energy [3]. Mpox spreads by close contact with an animal infected with the mpox virus; however, human-to-human transmission is also possible through skin-to-skin contact, respiratory droplets, oral fluids, or by contact with fabrics, objects, or surfaces contaminated with mpox virus [4].

Prior to May 2022, human mpox was an epidemic in African countries [3]. However, at the beginning of May 2022, after a confirmed case of mpox in an individual who returned from Nigeria to the United Kingdom was reported to the World Health Organization (WHO), subsequent clusters of mpox virus infections occurred in multiple non-epidemic countries [4,5]. As of 31 January 2023, a total of 85,469 confirmed cases of mpox across 110 countries or territories were reported globally [6]. The 2022 multiple-country mpox outbreak may be different from outbreaks before 2022 in epidemiology features. Our previous meta-analysis reported that the average age and comorbidity rate of mpox cases in 2022 was 35.52 years and 15.7%, respectively; both of them were significantly higher than those of cases before 2022 [7]. Compared with the rarely reported men who have sex with men (MSM) population in previous studies, the reported population in 2022 had a higher proportion of MSM (79.8%, 95% CI [65.5%, 94.2%]) [7,8]. In addition, studies have shown that compared with before 2022, the leading site of rash changed from cheek to genital mucosa [8]. Considering the above differences between the monkeypox epidemic in 2022 and that of before 2022, the mpox epidemic drew attention. The International Health Regulations Emergency Committee officially announced on 23 July 2022, that the mpox epidemic constitutes a Public Health Emergency of International Concern (PHEIC) [9].

With the development of the global economy, outbreaks of infectious diseases usually result in a public reaction in numerous countries other than the outbreak country. Few studies analyzed the public concerns about monkeypox recently. Thakur et al. found the neutral sentiment of mpox was present in most of the Tweets from 7 May 2022 and 23 July 2022 by performing sentiment analysis of the Tweets [10]. Sv et al. also reported that proportions of neutral sentiment (48.16%), positive sentiments (28.82%), and negative sentiments (23.01%), then described specific positive sentiments and negative sentiments in a text analytics study based on natural language processing [11]. Increasing search interest about a particular disease represents the development of an epidemic and limited knowledge of the disease [12]; therefore, it can be used to provide clinical epidemiologists with timely alerts of disease outbreaks or changes in treatment regimens with much earlier response than traditional health epidemiology [13]. Online search activity reflects matters of concern in a population, and the Google Trend Index (GTI) is a widely used indicator to observe public reactions and predict the trend of an epidemic [14,15,16,17,18]. However, a study analyzing the trend of online search activity in Google and its associations with the mpox epidemic at the global and national levels is lacking. Liu et al. simply described online activities (using Google Trends and Reddit) across US states and listed search interest scores for the top five most cases states [19]. Martins-Filho et al. mainly observed the online interest in mpox (using Google Trends) by using a line graph [20]. We introduced the GTI into the study to address the noteworthy association between online search activity and the mpox epidemic at global and national levels under different time lag days and supplemented its influencing factors, including socio-demographic characteristics and human resources for health levels. In addition, we explored the effect of PHEIC on changes in online search activity using time-series interrupted analysis to identify the differences and provide a reference on comprehensive management.

## 2. Materials and Methods

### 2.1. Daily New Cases of Mpox and Online Search Activity

Data on daily new cases of mpox from 1 May 2022 to 9 October 2022 were retrieved from Our World in Data (https://ourworldindata.org/monkeypox (accessed on 1 December 2022)). The Google Trends tool was utilized to retrieve data on internet patron search activity in the context of mpox. The Google Trends tool empowers researchers to study patterns and tendencies in Google search queries [21]. Here, online search activity was obtained using the keyword “monkeypox,” in this study. Online search activity was expressed as relative normalized search volume numbers (RNSNs). RNSNs range from 0 to 100 and reflect how many searches are performed for a keyword relative to the total number of searches on the internet over time. A value of 100 represents the time point at which the search term has reached its peak in popularity [20]. RNSNs were extracted day by day from 1 May to 9 October 2022. Further information on Google Trends is available at the relevant help pages (https://support.google.com/trends/ (accessed on 1 December 2022)).

### 2.2. Covariates

The data on demographic characteristics, including total population, population density, average years of schooling (average number of years people aged 25+ participated in formal education), socio-economic status (Gross Domestic Product (GDP) per capita 2021 (measured in constant 2017 international-$)), and public tourism were mainly collected from United Nations (UN) (http://data.un.org/ (accessed on 1 December 2022)) and World Bank (https://data.worldbank.org/ (accessed on 1 December 2022)). The UN’s public tourism and transport database was used to compile tourist/visitor arrivals to reflect population mobility. Data on health status, including HIV incidence and prevalence among 15–49-year-old individuals, rate of no access to handwashing facilities, and unsafe sanitation were obtained from the GBD Study 2019. We also extracted health workforce densities per 10,000 employed individuals for health workers by country and territory from the GBD Study 2019 [22].

### 2.3. Statistical Analysis

We used interrupted time-series analysis (ITS) to estimate the impact of the declaration of PHEIC on online search activity after adjusting for daily new cases among 194 countries or territories. The daily online search activity from 1 May to 9 October 2022 was the dependent variable (Y), and the declaration of IPEH from 23 July 2022 was the intervention cutoff point. The independent variables were X1, X2, and X3. The time variable (X1, counting variable) had values from 1 to 162, representing each day from 1 May to 9 October 2022. X2 referred to the declaration of IPEH, and we classified 162 days into the following two periods: the pre-PHEIC period (from 1 May 2022 to 23 July 2022) and the period after the declaration of PHEIC (from 24 July 2022 to 9 October 2022); the value of 0 was given to the time before the PHEIC declaration, and the value of 1 was given to the time after the PHEIC declaration (since 23 July 2022). X3 represented the slope (0—referring to the time before the PHEIC declaration (including 23 July 2022); 1 to 77—referring to each day after the PHEIC declaration (starting from 24 July 2022)). The values of all independent variables were taken day by day with even intervals. The fitted level and slope change model was determined as follows [23,24,25]:$$Y=\beta_{0}+\beta_{1}{\;*\;X}_{1}+\beta_{2}{\;*\;X}_{2}+\beta_{3}{\;*\;X}_{3}+\varepsilon ,$$
where $\beta_{1}$ was the slope of online search activity before the declaration, reflecting the daily average change trend in online search activity before the declaration of PHEIC; $\beta_{2}$ was the change level, reflecting the change in online search activity after the declaration of PHEIC; $\beta_{3}$ was the slope of change, and $\beta_{1}+\beta_{3}$ meant the slope after the declaration, reflecting the daily average change trend in online search activity after the declaration of PHEIC; and ε was error [23,24,25]. Then, we extracted the values of $\beta_{1}$, $\beta_{2}$, and $\beta_{1}+\beta_{3}$ to explore the influencing factors using a general linear regression (GLM) model.

Spearman correlation coefficient (rs) was measured to test the time-lag correlations between online search activity and daily new cases by −21, −14, −7, 0, +7, +14, and +21 days of lag. The positive lags corresponded to the time-lag effect of online search activity on daily new cases; in contrast, the negative lags represented the time-lag effect of daily new cases on online search activity among 102 countries or territories. A meta-analysis was conducted to show 95% confidence intervals (CIs) for the spearman correlation coefficients and the combined correlation coefficient for negative lags (−7, −14, −21), 0, and positive lags (7, 14, 21) to observe the time-lag effect of daily new cases on online search activity and the time-lag effect of online search activity on daily new cases [26]. Heterogeneity between the studies was evaluated by the *I*^2^ statistic, which denoted the total variation explained by the variation among the studies [27]. A random-effects model was adopted if significant heterogeneity existed between the studies (*I*^2^ ≥ 50%); otherwise, a fixed-effects model was applied (*I*^2^ < 50%) [27,28]. Then, we extracted the specific value of pooled Spearman correlation coefficients to explore the influencing factors using the GLM model on the relationship between online search activity and daily new cases. The analysis was conducted using R (version 4.1.0).

## 3. Results

### 3.1. The Trend in Global Online Search Activity

From 1 May to 9 October 2022, Global online search activity peaked on 23 May and 4 August 2022 (Figure 1).

Mpox was announced as a PHEIC on 23 July 2022 [9]. Before 23 July, there was a stable trend in global online search activity ($\beta_{1}=-0.164,\;P=0.062$); of 194 countries or territories, six countries or territories (6/194, 3.09%) had an increasing trend (e.g., Barbados, Bermuda, Brazil) and 42 (42/194, 21.65%) had a decreasing trend (e.g., Australia, Austria, Bangladesh) (Figure 2A and Appendix A). After the declaration of PHEIC, the global online search activity increased by 42.845% ($\beta_{2}=42.845,\;P<0.001$). The online search activity increased among 62 (62/194, 31.96%) countries or territories; among them, the number of countries or territories in Africa, Asia, Europe, North America, Oceania, and South America was four, 18, 22, 11, 2, and 5. The proportions of countries or territories with increasing change in Africa, Asia, Europe, North America, Oceania, and South America were 8.16% (4/49), 39.13% (18/46), 55.16% (22/43), 35.48% (11/31), 15.38% (2/13), and 41.67% (5/12), respectively. Brazil ($\beta_{2}=50.611,\;P<0.001$), United States ($\beta_{2}=$ 45.423, P<0.001), and Indonesia ($\beta_{2}=$ 40.298, P<0.001) ranked as the top three (Figure 2B and Appendix A).

In the period after the declaration of PHEIC (24 July 2022–9 October 2022), the slope of global online search activity was −0.164 + (−0.631) = −0.795, showing a downward trend ($\beta_{3}=-$0.631, P<0.001); similarly, 29 countries or territories had a downward trend, which were mainly in Asia (11/29, 37.93%) (Figure 2C and Appendix A). The proportions of countries or territories with a downward trend in Africa, Asia, Europe, North America, Oceania, and South America were 4.08% (2/49), 23.91% (11/46), 6.98% (3/43), 25.81% (8/31), 15.38% (2/13), and 25.00% (3/12), respectively. 

The trend before and after July 23 and the change in online search activity was affected by the rate of unsafe sanitation. When it increased by 1 per 100,000, the value of the trend before July 23 increased by 0.15% (95% CI: 0.03, 0.27); the value of the trend after July 23 increased by 0.30% (95% CI: 0.12, 0.48); and the value of change decreased by 14.86% (95% CI: −25.64, −4.08). Hence, countries or territories with a higher rate of unsafe sanitation were more likely to have a larger increase in the trend of online search activity before 23 July and after 23 July, and a smaller change in online search activity after 23 July 2022. 

### 3.2. Time-Lag Correlations of Online Search Activity and New Cases of Mpox

As shown in Appendix A, based on the overall global data, a negative correlation (*P* < 0.05) was observed when the lag was −21 days. Hence, the more daily new cases, the relatively lower the global online search activity would be observed in the next 21 days. Importantly, when the lag was 0, 7, 14, and 21 days, there were positive correlations between the two; hence, the higher the global online search activity, the more daily cases would be observed thereafter, and the strongest Spearman correlation was on days 7 and 14 (rs = 0.253) (Appendix A). Among 102 countries or territories, the proportion of countries or territories with positive correlations between online search activity and mpox new cases on days 0, 7, 14, and 21 was 20.59% (21/102), 12.75% (13/102), 11.76% (12/102), and 8.82% (9/102) (Figure 3).

There were positive correlations on days −21, −14, −7, 0, 7, 14, and 21 days in Brazil, where the highest correlation was on day 21 (rs = 0.467) (Appendix A).

We performed a meta-analysis and merged the correlations by positive and negative lags for the 102 countries or territories (Table 1).

The time-lag effect of daily new cases on global online search activity (negative lags: rs = −0.099; 95% CI: −0.193, −0.005; *P* = 0.038), and the time-lag effect of global online search activity on daily new cases was significant (positive lags: rs = 0.241; 95% CI: 0.147, 0.335; *P* < 0.001). As shown in Figure 4A and Table 1, there were eight countries or territories with a significant time-lag effect of global online search activity on daily new cases; the top three countries or territories were Brazil (positive lags: rs = 0.460; 95% CI: 0.366, 0.554; *P* < 0.001), United States (positive lags: rs = 0.244; 95% CI: 0.150, 0.337; *P* < 0.001), and Canada (positive lags: rs = 0.238; 95% CI: 0.144, 0.332; *P* < 0.001). As shown in Figure 4B and Table 1, there were nine countries or territories with a significant time-lag effect of daily new cases on global online search activity; the top three countries were Brazil (negative lags: rs = 0.244; 95% CI: 0.150, 0.338; *P* < 0.001), United Kingdom (negative lags: rs = 0.117; 95% CI: 0.024, 0.211; *P* < 0.05), and United Arab Emirates (negative lags: rs = 0.119; 95% CI: 0.025, 0.213; *P* < 0.05). 

The time-lag effect of daily new cases on global online search activity was affected by the average years of schooling; namely, when the average years of schooling increased by one, the Spearman correlation coefficient decreased by 1.05% (95% CI: −1.97, −0.12), which means that the daily new cases of countries or territories with higher average years of schooling were less likely to correlate with online search activity.

## 4. Discussion

To our knowledge, this is the first study to assess the trend and influencing factors of online search activity on mpox and explore the relationship between online search activity and daily new cases of mpox.

We found that before the declaration of PHEIC, there was a stable trend of global online search activity, and only 3.09% of countries or territories had an increasing trend. After the declaration of PHEIC, global online search activity increased by 42.845%. Some potential reasons may be associated with the increasing interest in mpox after the declaration of PHEIC. The authority of information sources may influence people’s attention to infectious epidemics. A text analytics study showed that from 1 June 2022 to 25 June 2022, the general public did not panic to a significant extent about the mpox virus [11]. Compared with numerous information from social media in the early stage, the progression released by the WHO seemed to be more reliable and attractive. Further, the severity of the diseases also affected the general public’s concerns [11]. The PHEIC showing increasing severity of the mpox epidemic resulted in the perception of severity and concerns in mpox for people increased. In addition, the declaration of PHEIC may increase the number of guidelines or recommendations from governments or policy-making bodies, and these guidelines or recommendations may heat online discussion [10]. However, further research is still needed to support the above possible reasons. Although the proportion of countries or territories with increasing changes in online search activity was 31.96% (62/194), the proportions in Africa and Oceania were only 8.16% and 15.38%, respectively. There is no doubt that the declaration of PHEIC increased people’s interest in mpox; however, interest increased only in some of the countries, especially in epidemic countries. Interest in searching for mpox is an indicator of behavioral information. People in epidemic countries may search for the latest knowledge to reduce the risk of mpox virus infection, compared with other countries. Other parts of our findings supported this speculation. We found that the time-lag effect of daily new cases on online search activity was larger in Brazil and the United Kingdom—countries with a severe mpox epidemic. The time-lag effect of daily new cases on online search activity was attenuated in countries or territories with higher average years of schooling.

Subsequently, in the period after the declaration of PHEIC, global online search activity showed a downward trend, possibly due to the relatively stable number of new cases globally. The proportions of countries or territories with a downward trend in Asia, North America, and South America were all higher than 20%. However, the decline of interest in mpox reminded us that measures should be taken to construct trusted sources for supplementing reliable information. If people lack knowledge of the transmission of the virus and the prevention measures, they seek information more frequently; meanwhile, the possibilities of finding inaccurate information are higher, and the outbreak exacerbates [20]. We observed that countries or territories with a higher rate of unsafe sanitation were more likely to have a larger increase in the trend of online search activity before 23 July and after 23 July and a smaller change in online search activity after 23 July 2022. This highlights the crucial characteristic of countries or territories that need the construction of trusted sources.

The other important finding was the prediction ability of online search activity on new cases. We found that the time-lag effect of global online search activity on new cases daily was significant. The top three countries with a higher effect were Brazil, the United States, and Canada. Recently, web-based tools, including social media and search engines, have already provided an opportunity to discover new and undiagnosed diseases [29]. Our findings supported the use of the Google Trend index as a useful tool to monitor the spread of infectious diseases, especially in epidemic regions. However, our results also indicated that different relationships existed when time lags changed; namely, the proportion of countries or territories with positive correlations between online search activity and mpox new cases was lower when time lag days increased, so Google Trends cannot be used for precise real-time epidemiological surveillance of mpox, especially for specific regions. However, Google Trends may still be a considerable tool that helps governments and researchers capture signals of an epidemic, formulate prevention and control plans, and promote public knowledge. The above findings indicate that this monitoring tool should be dynamic and adjusted according to the epidemic situation; setting different time lag days to find precise time lag days to predict the spread of infectious diseases was necessary. Thakur et al. found that neutral sentiment was present in most of the Tweets, and followed by negative and positive sentiments from 7 May 2022 and 23 July 2022 [10]. One study reported that the negative sentiments of mpox discussed by ordinary people among the tweets mainly included the deaths, the severity, lesions, transmissions, vaccines and safety to travel, etc., in the early stage [11]. Therefore, if the monitoring tool detected the online search activity in advance, understood the general public’s concerns, and found and corrected the false information, controlling the epidemic timely and effectively would be possible. In addition, specific policies recommended by governments or policy-making bodies might be disagreed or agreed with online, allowing governing bodies to reassess their policies in real-time to address issues with their handling of the crisis [10]. 

Some limitations of this study should be demonstrated. First, due to the restriction of language, the Google Trends Index may underestimate online search activity in some countries. Second, Google Trends presented a relative volume that meant that we could not compare the internet attention behavior between countries directly. Third, data on internet attention behavior on mpox were unavailable before 2022, so we could not compare the trends in 2022 with other years.

## 5. Conclusions

In conclusion, the changes in online search activity after the declaration of PHEIC showed interest behavior in mpox was insufficient, especially in Africa. The decreasing trend of online search activity after the declaration of PHEIC in North America should be paid attention to. Based on the significant time-lag effect of global online search activity on new cases daily, online search activity could be used to correlate and predict the outbreak in epidemic countries and worldwide. It is desirable to construct a reliable information source for the public and use the collected online search activity to capture signals of an epidemic, formulate prevention and control plans, and promote public knowledge.

## Figures and Tables

**Figure 1 ijerph-20-03395-f001:**
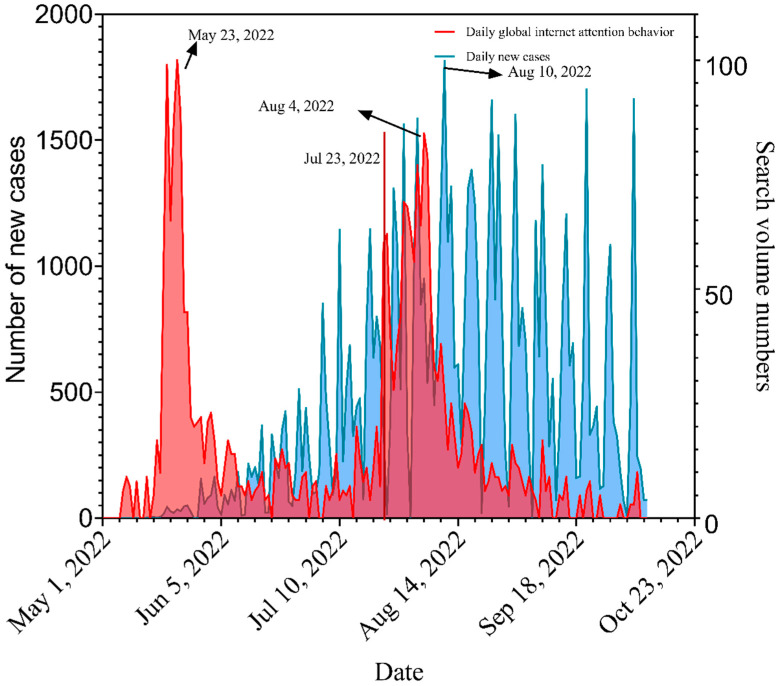
Worldwide daily online search activity and mpox new cases.

**Figure 2 ijerph-20-03395-f002:**
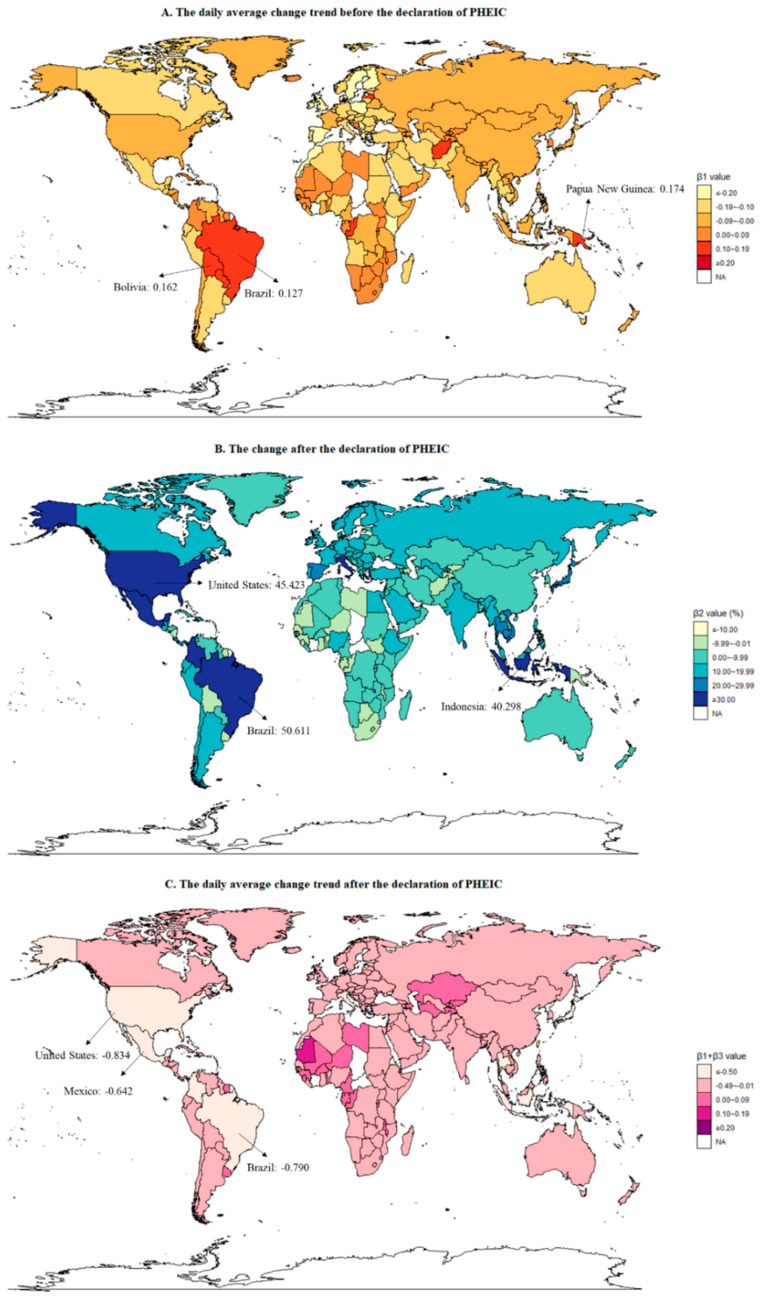
The trend and change in online search activity before and after public health emergency of international concern: (**A**) the daily average change trend of online search activity before the declaration of PHEIC; (**B**) the change of online search activity after the declaration of PHEIC; (**C**) the daily average change trend of online search activity after the declaration of PHEIC.

**Figure 3 ijerph-20-03395-f003:**
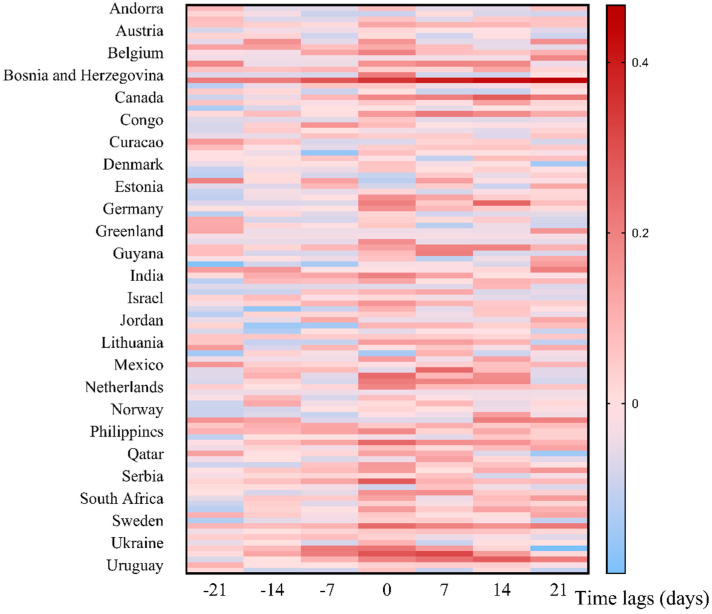
Heat plot of the time lag correlation of the online search activity and mpox new cases in 102 countries or territories.

**Figure 4 ijerph-20-03395-f004:**
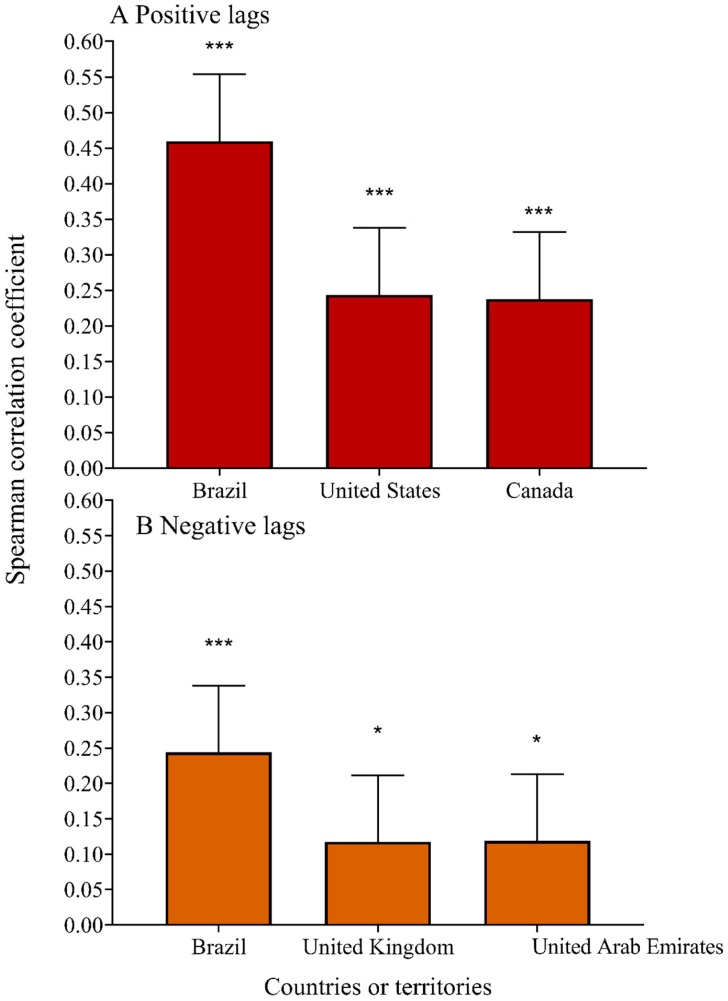
Time-lag correlations of online search activity and daily new cases in the top three countries or territories by positive and negative lags. * *P* < 0.05; *** *P* < 0.001.

**Table 1 ijerph-20-03395-t001:** Time-lag correlations of online search activity and daily new cases in 102 countries or territories by positive and negative lags.

	Negative Lags		Positive Lags	
Countries or Territories	Spearman Correlation Coefficient	*P*-Value	Spearman Correlation Coefficient	*P*-Value
World	−0.099 (−0.193, −0.005)	0.038	0.241 (0.147, 0.335)	<0.001
Andorra	−0.012 (−0.106, 0.082)	0.795	−0.019 (−0.113, 0.075)	0.692
Argentina	−0.035 (−0.129, 0.059)	0.469	−0.046 (−0.140, 0.048)	0.340
Aruba	−0.021 (−0.115, 0.073)	0.662	0.013 (−0.081, 0.107)	0.787
Australia	0.037 (−0.057, 0.131)	0.437	0.081 (−0.013, 0.175)	0.092
Austria	−0.051 (−0.145, 0.042)	0.283	−0.025 (−0.119, 0.069)	0.600
Bahamas	−0.011 (−0.105, 0.083)	0.819	−0.055 (−0.149, 0.039)	0.248
Bahrain	0.020 (−0.126, 0.166)	0.790	0.021 (−0.128, 0.169)	0.785
Barbados	0.112 (0.018, 0.206)	0.019	−0.008 (−0.102, 0.086)	0.863
Belgium	0.025 (−0.069, 0.119)	0.598	0.077 (−0.017, 0.171)	0.107
Benin	−0.036 (−0.130, 0.058)	0.451	0.039 (−0.107, 0.185)	0.601
Bermuda	0.034 (−0.117, 0.185)	0.662	0.113 (−0.038, 0.264)	0.141
Bolivia	0.073 (−0.021, 0.167)	0.127	0.056 (−0.038, 0.149)	0.247
Bosnia and Herzegovina	−0.069 (−0.163, 0.025)	0.150	−0.058 (−0.152, 0.036)	0.226
Brazil	0.244 (0.150, 0.338)	<0.001	0.460 (0.366, 0.554)	<0.001
Bulgaria	−0.032 (−0.126, 0.062)	0.505	−0.016 (−0.110, 0.078)	0.742
Cameroon	−0.013 (−0.107, 0.081)	0.789	−0.067 (−0.160, 0.027)	0.165
Canada	−0.013 (−0.107, 0.081)	0.791	0.238 (0.144, 0.332)	<0.001
Chile	0.016 (−0.078, 0.110)	0.732	0.046 (−0.048, 0.140)	0.335
China	−0.068 (−0.162, 0.026)	0.155	−0.017 (−0.111, 0.076)	0.715
Colombia	0.031 (−0.063, 0.125)	0.513	0.173 (0.079, 0.267)	<0.001
Congo	−0.062 (−0.156, 0.032)	0.196	−0.056 (−0.150, 0.038)	0.244
Costa Rica	0.042 (−0.094, 0.178)	0.548	−0.018 (−0.112, 0.076)	0.707
Croatia	−0.010 (−0.104, 0.084)	0.836	−0.026 (−0.120, 0.068)	0.585
Cuba	−0.008 (−0.102, 0.086)	0.867	0.012 (−0.082, 0.106)	0.798
Curacao	0.044 (−0.050, 0.138)	0.361	−0.002 (−0.096, 0.092)	0.961
Cyprus	0.000 (−0.094, 0.094)	0.994	0.054 (−0.040, 0.147)	0.264
Czechia	−0.076 (−0.170, 0.018)	0.115	−0.009 (−0.103, 0.085)	0.847
Democratic Republic of Congo	0.013 (−0.081, 0.107)	0.786	0.009 (−0.085, 0.102)	0.859
Denmark	−0.018 (−0.112, 0.076)	0.701	−0.057 (−0.151, 0.037)	0.234
Dominican Republic	−0.062 (−0.156, 0.032)	0.196	0.007 (−0.087, 0.101)	0.888
Ecuador	−0.060 (−0.154, 0.034)	0.209	0.012 (−0.082, 0.106)	0.803
El Salvador	0.113 (0.019, 0.207)	0.018	0.034 (−0.060, 0.128)	0.477
Estonia	−0.011 (−0.105, 0.083)	0.821	0.023 (−0.071, 0.117)	0.627
Finland	−0.059 (−0.153, 0.035)	0.215	−0.028 (−0.122, 0.066)	0.554
France	−0.051 (−0.145, 0.043)	0.284	0.066 (−0.028, 0.160)	0.167
Georgia	−0.066 (−0.160, 0.028)	0.166	0.126 (0.032, 0.220)	0.009
Germany	−0.011 (−0.105, 0.083)	0.814	0.048 (−0.046, 0.142)	0.317
Ghana	−0.051 (−0.145, 0.043)	0.288	−0.065 (−0.159, 0.029)	0.174
Gibraltar	−0.015 (−0.109, 0.079)	0.753	−0.006 (−0.100, 0.088)	0.903
Greece	0.040 (−0.054, 0.134)	0.401	−0.083 (−0.176, 0.011)	0.085
Greenland	0.010 (−0.084, 0.104)	0.833	0.023 (−0.071, 0.117)	0.633
Guadeloupe	−0.034 (−0.128, 0.060)	0.482	−0.031 (−0.125, 0.063)	0.520
Guam	−0.052 (−0.146, 0.042)	0.282	−0.050 (−0.144, 0.044)	0.300
Guatemala	0.061 (−0.033, 0.155)	0.202	0.169 (0.075, 0.263)	<0.001
Guyana	−0.026 (−0.120, 0.068)	0.589	0.023 (−0.164, 0.210)	0.807
Honduras	−0.028 (−0.122, 0.066)	0.556	−0.012 (−0.106, 0.082)	0.805
Hungary	−0.137 (−0.230, −0.043)	0.004	0.017 (−0.077, 0.111)	0.725
Iceland	0.093 (−0.001, 0.187)	0.052	0.067 (−0.068, 0.202)	0.331
India	0.085 (−0.008, 0.179)	0.075	0.037 (−0.057, 0.131)	0.440
Indonesia	0.015 (−0.079, 0.109)	0.754	0.059 (−0.035, 0.153)	0.217
Iran	−0.065 (−0.159, 0.029)	0.177	−0.014 (−0.108, 0.080)	0.773
Ireland	−0.043 (−0.137, 0.051)	0.365	0.002 (−0.092, 0.096)	0.967
Israel	0.029 (−0.065, 0.123)	0.548	−0.052 (−0.146, 0.042)	0.277
Italy	0.033 (−0.061, 0.127)	0.495	0.062 (−0.032, 0.156)	0.194
Jamaica	−0.113 (−0.207, −0.019)	0.018	−0.005 (−0.099, 0.089)	0.924
Japan	−0.020 (−0.114, 0.074)	0.676	−0.012 (−0.106, 0.082)	0.798
Jordan	0.009 (−0.085, 0.103)	0.856	0.005 (−0.089, 0.099)	0.915
Latvia	−0.094 (−0.188, 0.000)	0.051	0.065 (−0.029, 0.159)	0.176
Lebanon	−0.064 (−0.158, 0.030)	0.182	−0.038 (−0.132, 0.056)	0.433
Liberia	0.052 (−0.042, 0.146)	0.275	0.041 (−0.053, 0.135)	0.393
Lithuania	−0.051 (−0.145, 0.043)	0.288	0.045 (−0.101, 0.191)	0.548
Luxembourg	0.052 (−0.042, 0.146)	0.281	0.043 (−0.051, 0.137)	0.371
Malta	−0.045 (−0.139, 0.049)	0.349	−0.005 (−0.099, 0.089)	0.911
Martinique	−0.039 (−0.133, 0.055)	0.419	0.017 (−0.077, 0.111)	0.726
Mexico	0.071 (−0.023, 0.165)	0.137	0.066 (−0.028, 0.160)	0.171
Moldova	0.035 (−0.059, 0.129)	0.470	0.092 (−0.087, 0.271)	0.315
Montenegro	−0.007 (−0.101, 0.087)	0.887	0.069 (−0.068, 0.206)	0.326
Morocco	−0.039 (−0.133, 0.055)	0.417	0.102 (−0.074, 0.278)	0.257
Netherlands	0.021 (−0.073, 0.115)	0.665	0.069 (−0.025, 0.163)	0.150
New Caledonia	−0.042 (−0.136, 0.052)	0.377	−0.040 (−0.134, 0.054)	0.403
New Zealand	−0.007 (−0.100, 0.087)	0.892	−0.027 (−0.121, 0.067)	0.573
Nigeria	−0.001 (−0.095, 0.093)	0.981	0.018 (−0.076, 0.112)	0.710
Norway	−0.057 (−0.151, 0.037)	0.238	−0.021 (−0.115, 0.073)	0.667
Panama	−0.085 (−0.179, 0.009)	0.075	0.021 (−0.073, 0.115)	0.656
Paraguay	0.075 (−0.069, 0.218)	0.307	0.114 (−0.065, 0.292)	0.213
Peru	0.094 (0.000, 0.188)	0.049	0.078 (−0.016, 0.172)	0.105
Philippines	0.106 (0.012, 0.200)	0.026	0.061 (−0.033, 0.155)	0.204
Poland	−0.036 (−0.130, 0.058)	0.448	0.008 (−0.086, 0.102)	0.867
Portugal	0.061 (−0.033, 0.155)	0.200	0.147 (0.053, 0.241)	0.002
Puerto Rico	−0.018 (−0.112, 0.076)	0.711	0.072 (−0.022, 0.166)	0.134
Qatar	0.046 (−0.048, 0.140)	0.341	−0.036 (−0.185, 0.113)	0.636
Romania	−0.037 (−0.131, 0.057)	0.437	0.033 (−0.061, 0.127)	0.494
Russia	−0.033 (−0.127, 0.060)	0.485	0.037 (−0.057, 0.131)	0.436
Saudi Arabia	0.042 (−0.052, 0.136)	0.376	0.087 (−0.007, 0.181)	0.069
Serbia	0.004 (−0.090, 0.098)	0.934	−0.014 (−0.108, 0.080)	0.772
Singapore	0.076 (−0.018, 0.170)	0.112	0.085 (−0.009, 0.179)	0.077
Slovakia	−0.006 (−0.100, 0.088)	0.903	−0.011 (−0.104, 0.083)	0.826
Slovenia	−0.046 (−0.140, 0.048)	0.339	0.089 (−0.005, 0.183)	0.063
South Africa	0.016 (−0.078, 0.110)	0.738	0.058 (−0.035, 0.152)	0.223
South Korea	−0.038 (−0.132, 0.056)	0.433	0.017 (−0.077, 0.111)	0.726
Spain	−0.031 (−0.125, 0.063)	0.514	0.090 (−0.004, 0.184)	0.060
Sudan	0.037 (−0.057, 0.131)	0.443	0.037 (−0.057, 0.131)	0.441
Sweden	−0.073 (−0.167, 0.021)	0.129	−0.026 (−0.120, 0.068)	0.585
Switzerland	0.089 (−0.005, 0.183)	0.065	0.194 (0.100, 0.288)	<0.001
Thailand	−0.004 (−0.098, 0.090)	0.936	0.051 (−0.043, 0.145)	0.284
Turkey	0.055 (−0.039, 0.149)	0.254	−0.045 (−0.139, 0.049)	0.352
Ukraine	−0.044 (−0.138, 0.050)	0.361	−0.034 (−0.128, 0.060)	0.476
United Arab Emirates	0.119 (0.025, 0.213)	0.013	−0.009 (−0.208, 0.190)	0.931
United Kingdom	0.117 (0.024, 0.211)	0.014	0.165 (−0.010, 0.340)	0.064
United States	0.011 (−0.083, 0.105)	0.811	0.244 (0.150, 0.337)	<0.001
Uruguay	0.028 (−0.066, 0.121)	0.566	0.020 (−0.074, 0.114)	0.682
Venezuela	−0.056 (−0.150, 0.038)	0.245	−0.063 (−0.157, 0.031)	0.190

## Data Availability

Our data were sourced from the database of Our World in Data (https://ourworldindata.org/monkeypox (accessed on 1 December 2022)), Google Trends (https://support.google.com/trends/ (accessed on 1 December 2022)), United Nations (UN) (http://data.un.org/ (accessed on 1 December 2022)) and World Bank (https://data.worldbank.org/ (accessed on 1 December 2022)) and GBD Study 2019.

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
