# Peer review of "Trends in Online Search Activity and the Correlation with Daily New Cases of Monkeypox among 102 Countries or Territories"

_ijerph, 2023, doi:10.3390/ijerph20043395_

Round 1

Reviewer 1 Report

The paper is titled - “Trends in online search activity and the correlation with daily new cases of monkeypox among 102 countries or territories”. The authors obtained data on the daily new cases and online search activity of monkeypox from Our World in Data and Google Trends tool. Segmented interrupted time-series analysis was used by the authors to estimate the impact of declaration of Public Health Emergency of International Concern (PHEIC) from July 23, 2022, on online search activity, after adjusting for daily new cases. Spearman correlation coefficient was thereafter measured in this work to test the time-lag correlations between online search activity and daily new cases. A meta-analysis was conducted by the authors to show 95% confidence intervals (CIs) for the Spearman correlation coefficients. The factors influencing the trend in global online search activity and the correlation with monkeypox epidemic were finally analyzed in this work using a general linear regression (GLM) model. The work seems novel. However, the presentation of the paper needs improvement in multiple parts. It is suggested that the authors make the necessary changes/updates to their paper as per the following comments:

  1. The abstract is too long. Please refer to the guidelines for the abstract at this link: https://www.mdpi.com/journal/ijerph/instructions. In the Microsoft Word submission template, it is stated - “Abstract: A single paragraph of about 200 words maximum.”

  2. This is a very similar paper that used Google Trends in the context of monkeypox https://ieeexplore.ieee.org/abstract/document/9983632. Please state how the work proposed in this paper is different from this paper.

  3. In Section 4, the authors have stated the specific findings such as “We found that before the declaration of PHEIC, there was a stable trend of global online search activity, and only 3.09% countries or territories had an increasing trend. After the declaration of PHEIC, the global online search activity increased by 42.845%”. These are novel findings. However, please present potential reasons or discuss a list of possible circumstances that may have contributed to these results. 

  4. The Introduction section is very short and written like a conference paper. A proper literature review is also missing. The authors have not reviewed any of the recent works related to R&D focused on monkeypox. Please add a literature review section and review some of the recent works that have focused on monkeypox such as https://doi.org/10.3390/idr14060087 and https://doi.org/10.1016%2Fj.tmaid.2022.102404

  5. In Section 3.1 the authors state - “From May 1 to October 9, 2022, there were two peaks of worldwide interest……”. However, in Figure 1, the data is provided for October 21. As these dates don’t match please state why different dates were used for the analysis.

  6. Minor Comments:

    1. The numbering of references should be corrected. There are two numbers before each reference

    2. Reference [7] seems to be incomplete. Is this a conference paper or a journal paper?

    3. The caption for Figure 2 should be after Figure 2 and not on the next page

Author Response

Reviewer(s)' Comments to Author:

Reviewer: 1

The paper is titled - “Trends in online search activity and the correlation with daily new cases of monkeypox among 102 countries or territories”. The authors obtained data on the daily new cases and online search activity of monkeypox from Our World in Data and Google Trends tool. Segmented interrupted time-series analysis was used by the authors to estimate the impact of declaration of Public Health Emergency of International Concern (PHEIC) from July 23, 2022, on online search activity, after adjusting for daily new cases. Spearman correlation coefficient was thereafter measured in this work to test the time-lag correlations between online search activity and daily new cases. A meta-analysis was conducted by the authors to show 95% confidence intervals (CIs) for the Spearman correlation coefficients. The factors influencing the trend in global online search activity and the correlation with monkeypox epidemic were finally analyzed in this work using a general linear regression (GLM) model. The work seems novel. However, the presentation of the paper needs improvement in multiple parts. It is suggested that the authors make the necessary changes/updates to their paper as per the following comments:

Response: Thanks. We have revised this manuscript based on reviewers’ suggestions.

  1. The abstract is too long. Please refer to the guidelines for the abstract at this link: https://www.mdpi.com/journal/ijerph/instructions. In the Microsoft Word submission template, it is stated - “Abstract: A single paragraph of about 200 words maximum.”

Response: Thanks. We have restated our abstract based on the guidelines for the abstract and limited words of the abstract as shown in page 1-2 “Research assessing the trend in online search activity on monkeypox (mpox) and the correlation with mpox epidemic at global and national level is scarce. The trend of online search activity, and the time-lag correlations between it and daily new mpox cases were estimated by using seg-mented interrupted time-series analysis and spearman correlation coefficient (rs), respectively. We found that after the declaration of Public Health Emergency of International Concern (PHEIC), the proportion of countries or territories with increasing changes in online search activity were lowest in Africa (8.16%, 4/49), and that of with a downward trend in online search activity was highest in North America (8/31, 25.81%). The time-lag effect of global online search activity on daily new cases was significant (rs = 0.24). There were eight countries or territories with significant time-lag effect, the top three countries or territories were Brazil (rs = 0.46), United States (rs = 0.24), and Canada (rs = 0.24). Interest behavior in mpox was insufficient, even after the declaration of PHEIC, especially in Africa and North America. Online search activity could be used as an early indicator of the outbreak of mpox at global level and in epidemic countries.”. 

  1. This is a very similar paper that used Google Trends in the context of monkeypox https://ieeexplore.ieee.org/abstract/document/9983632. Please state how the work proposed in this paper is different from this paper.

Response: Thanks. After reading this paper https://ieeexplore.ieee.org/abstract/document/9983632, this paper only used the Google Trends of US by August 26, 2022, and it simply descript the the Google Trends across states. Compared with this paper, our study included the Google Trends at global level and more countries, furthermore, our study analyzed the trend of it. Importantly, our study additionally included the new monkeypox cases and analyze the correlation between it and the Google Trends.  We also added it in introduction as shown in page 3 “Liu et al. simply descripted online activities (using Google Trends and Reddit) across US states, and listed search interests scores for top five most cases states [19].”.

  1. In Section 4, the authors have stated the specific findings such as “We found that before the declaration of PHEIC, there was a stable trend of global online search activity, and only 3.09% countries or territories had an increasing trend. After the declaration of PHEIC, the global online search activity increased by 42.845%”. These are novel findings. However, please present potential reasons or discuss a list of possible circumstances that may have contributed to these results.

Response: Thanks. We have added potential reasons in discussion as shown in page 12 line 293-305“Some potential reasons may be associated with the increasing interest in mpox after the declaration of PHEIC. The authority of information source may influence people's attention on infectious epidemic. A text analytics study showed that from June 1, 2022 to June 25, 2022 the general public didn’t panick to much extent about the mpox virus [11]. Compared with numerous information from social media in early stage, the progression released by the WHO seemed to be more reliable and attractable. What’s more, the severity of the diseases also affected general public’s concerns [11]. The PHEIC showing increasing severity of mpox epidemic resulted in the perception of severity and concerns in mpox for people increased. In addition, the declaration of PHEIC may increase the number of guidelines or recommendations from governments or policy-making bodies, and these guidelines or recommendations may heat online discussion [10]. However, further research is still needed to support the above possible reasons.”.

  1. The Introduction section is very short and written like a conference paper. A proper literature review is also missing. The authors have not reviewed any of the recent works related to R&D focused on monkeypox. Please add a literature review section and review some of the recent works that have focused on monkeypox such as https://doi.org/10.3390/idr14060087 and https://doi.org/10.1016%2Fj.tmaid.2022.102404

Response: Thanks. We have added a literature review section and review some of the recent works that have focused on monkeypox as shown in page 2 line 87-99 “The 2022 multiple-country mpox outbreak may be different with outbreak before 2022 on epidemiology features. Our previous meta-analysis reported that the average age and comorbidity rate of mpox cases in the 2022 were 35.52 years and 15.7%, respectively, both of them were significantly higher than those of cases before 2022 [7]. Com-pared with the rarely reported men who have sex with men (MSM) population in previous studies, the reported population in 2022 had a higher proportion of MSM (79.8%, 95% CI [65.5%, 94.2%]) [7,8]. In addition, studies had shown that compared with before 2022, the leading site of rash changed from cheek to genital mucosa [8]. Considering the above differences between monkeypox epidemic in 2022 and that of before 2022, mpox epidemic drew the attention. The International Health Regulations Emergency Committee officially announced on July 23, 2022, that the mpox epidemic constitutes a Public Health Emergency of International Concern (PHEIC) [9]” and page 3 line 102-120 “Few studies analyzed the public concerns in monkeypox recently. Thakur et al. found the neutral sentiment of mpox was present in most of the Tweets from 7 May 2022 and 23 July 2022 by performing sentiment analysis of the Tweets [10]. Sv et al. also reported that proportions of neutral sentiment (48.16%), positive sentiments (28.82%) and negative sentiments (23.01%), then descripted specific positive sentiments and negative sentiments in a text analytics study based on Natural Language Processing [11]. Increasing search interest about a particular disease represents the development of an epi-demic and limited knowledge on the disease [12]; therefore, it can be used to provide clinical epidemiologists with timely alerts of disease outbreaks or changes in treatment regimens with much earlier response than traditional health epidemiology [13]. Online search activity reflects matters of concern in a population, and the Google Trend Index (GTI) is a widely used indicator to observe public reactions and predict the trend of an epidemic [14-18]. However, study analyzing the trend of online search activity in Google and its associations with mpox epidemic at global and national level is lacked. Liu et al. simply descripted online activities (using Google Trends and Reddit) across US states, and listed search interests scores for top five most cases states [19]. Martins-Filho et al. mainly observed the online interest in mpox (using Google Trends) by using a line graph [20]. We introduced the GTI into the study to address the noteworthy association between online search activity and the mpox epidemic at global and national level under different time lag days, and supplemented its influencing factors including socio-demographic characteristics and human resources for health levels. In addition, we explored the effect of PHEIC on changes in online search activity using time-series interrupted analysis to identify the differences and provide reference on comprehensive management.”.

  1. In Section 3.1 the authors state - “From May 1 to October 9, 2022, there were two peaks of worldwide interest……”. However, in Figure 1, the data is provided for October 21. As these dates don’t match please state why different dates were used for the analysis.

Response: Thanks. The unit measurement is not one day of Figure 1. We have confirmed that Figure 1 was consistent with the description.

  1. Minor Comments:

The numbering of references should be corrected. There are two numbers before each reference

Response: Thanks. We have deleted the duplicated number before reference and checked all references.

Reference [7] seems to be incomplete. Is this a conference paper or a journal paper?

Response: Thanks. We have checked all references and adjusted the reference 7 which was a web page.

The caption for Figure 2 should be after Figure 2 and not on the next page

Response: Thanks. We have adjusted it to make sure they are in the same page after this article the accepted tacked changes.

Reviewer 2 Report

This is a very interesting article and I think the readers of IJERPH will benefit from this publication. Here is some feedback to improve the quality of this manuscript.

1. The discussion of Lines 282 - 295 needs to be more specific and accurate so that the recommendation regarding the practical application could be more useful for the end-users. I consider that this is the most significant part of this manuscript, and the discussion/analysis is too brief and narrow.

2. More clarifications are needed regarding the timeline in the Discussion. For example, the time is "after the declaration of PHEIC" in Lines 256 -237, and it is "In the PHEIC period" in the Line of 270.  

3. A minor issue of academic presentation: the font size of Lines 81-83 is different from the rest of the lines.

Author Response

Reviewer: 2

This is a very interesting article and I think the readers of IJERPH will benefit from this publication. Here is some feedback to improve the quality of this manuscript.

Response: Thanks.

  1. The discussion of Lines 282 - 295 needs to be more specific and accurate so that the recommendation regarding the practical application could be more useful for the end-users. I consider that this is the most significant part of this manuscript, and the discussion/analysis is too brief and narrow.

Response: Thanks. We have supplemented the recommendation regarding the practical application as shown in page 13 line 343-358“Above findings indicated that this monitoring tool should be dynamic and adjusted according to the epidemic situation, setting different time lag days to find precise time lag days to predict the spread of infectious diseases was necessary. Thakur et al. found that neutral sentiment was present in most of the Tweets, and followed by negative and positive sentiments from 7 May 2022 and 23 July 2022 [10]. One study reported that the negative sentiments of mpox discussed by ordinary people among the tweets mainly included the deaths, the severity, lesions, transmissions, vaccines and safe to travel, etc, in the early stage [11]. Therefore, if monitoring tool detected the online search activity in advance, understood the general public’s concerns, found and corrected the false information, controlling epidemic timely and effectively was prospective. In addition, specific policies which recommended by governments or policy-making bodies might be disagreed or agreed online, revised inappropriate policies by capturing online search activity was also considerable to address crisis [10]”.

  1. More clarifications are needed regarding the timeline in the Discussion. For example, the time is "after the declaration of PHEIC" in Lines 256 -237, and it is "In the PHEIC period" in the Line of 270.

Response: Thanks. In order to emphasize the trend was the change during a period, we used "In the PHEIC period". In fact, “In the PHEIC period” is the period after the declaration of PHEIC, they are the same meanings, we have corrected it as “In the period after the declaration of PHEIC (July 24, 2022–October 9, 2022)".

  1. A minor issue of academic presentation: the font size of Lines 81-83 is different from the rest of the lines.

Response: Thanks. We have corrected the font size of Lines 81-83.

Round 2

Reviewer 1 Report

The authors have revised the paper as per all my comments and feedback. I do not have any additional comments at this point. I recommend the publication of the paper in its current form.